# In-context learning enables multimodal large language models to classify cancer pathology images

Dyke Ferber[1,2,3], Georg Wölflein [4], Isabella C. Wiest[3,5], Marta Ligero[3], Srividhya Sainath[3], Narmin Ghaffari Laleh [3], Omar S. M. El Nahhas[3], Gustav Müller-Franzes[6], Dirk Jäger[1,2], Daniel Truhn[6] & Jakob Nikolas Kather [1,2,3,7] ✉

Medical image classification requires labeled, task-specific datasets which are used to train deep learning networks de novo, or to fine-tune foundation models. However, this process is computationally and technically demanding. In language processing, in-context learning provides an alternative, where models learn from within prompts, bypassing the need for parameter updates. Yet, in-context learning remains underexplored in medical image analysis. Here, we systematically evaluate the model Generative Pretrained Transformer 4 with Vision capabilities (GPT-4V) on cancer image processing with in-context learning on three cancer histopathology tasks of high importance: Classification of tissue subtypes in colorectal cancer, colon polyp subtyping and breast tumor detection in lymph node sections. Our results show that in-context learning is sufficient to match or even outperform specialized neural networks trained for particular tasks, while only requiring a minimal number of samples. In summary, this study demonstrates that large vision language models trained on non-domain specific data can be applied out-of-the box to solve medical image-processing tasks in histopathology. This democratizes access of generalist AI models to medical experts without technical background especially for areas where annotated data is scarce.

Artificial intelligence (AI) is about to transform healthcare. While its potential is immense, it also presents unique challenges in medicine, arising from the field's inherent complexity and the critical need for accuracy and reliability[1]. Over the last years, applications of AI have been developed that focus on specific areas, especially computer vision models in radiology[2] and pathology[3], or skin cancer detection[4] for oncology.

Histopathology plays a central role in diagnosing diseases, notably cancer, and has consistently been at the forefront of computational advancements in medicine[5]. Recent developments have enabled the detection of cancer subtypes[6] and biomarkers like genetic alterations[7] which can potentially stratify and improve patient care directly from routine hematoxylin and eosin (H&E) stained microscopic images[7]. The current gold standard for computational pathology is training

[1]National Center for Tumor Diseases (NCT), Heidelberg University Hospital, Heidelberg, Germany. [2]Department of Medical Oncology, Heidelberg University Hospital, Heidelberg, Germany. [3]Else Kroener Fresenius Center for Digital Health, Technical University Dresden, Dresden, Germany. [4]School of Computer Science, University of St Andrews, St Andrews, UK. [5]Department of Medicine II, Medical Faculty Mannheim, Heidelberg University, Mannheim, Germany. [6]Department of Diagnostic and Interventional Radiology, University Hospital Aachen, Aachen, Germany. [7]Department of Medicine I, University Hospital Dresden, Dresden, Germany. ✉e-mail: jakob_nikolas.kather@tu-dresden.de

vision foundation models[8] based on a vast and diverse dataset of images that can easily be customized for clinically relevant applications[9,10]. However, these foundation models need a substantial volume of domain-specific images during training and are restricted to

vision applications only. Moreover, before being applied to a medical task, these models require an additional re-training stage (fine-tuning) that is in itself computationally demanding[11] and requires additional annotated training data. This last step needs to be repeated for every

**A**

| Dataset | CRC100K | PatchCamelyon | MHIST |
|---|---|---|---|
| Description | H&E patches from 86 WSIs of human colorectal cancer | Histologic images of lymph node sections with or without metastatic breast cancer | Microscopic images of colorectal polyps |
| # samples | 7180 | 32768 | 3152 |
| classes | ADI, BACK*, DEB, LYM, MUC, MUS, NORM, STR, TUM | Tumor, No Tumor | SSA, HP |
| # eval binary | 60 | 60 | 60 |
| complete | 120 | – | – |

**B**

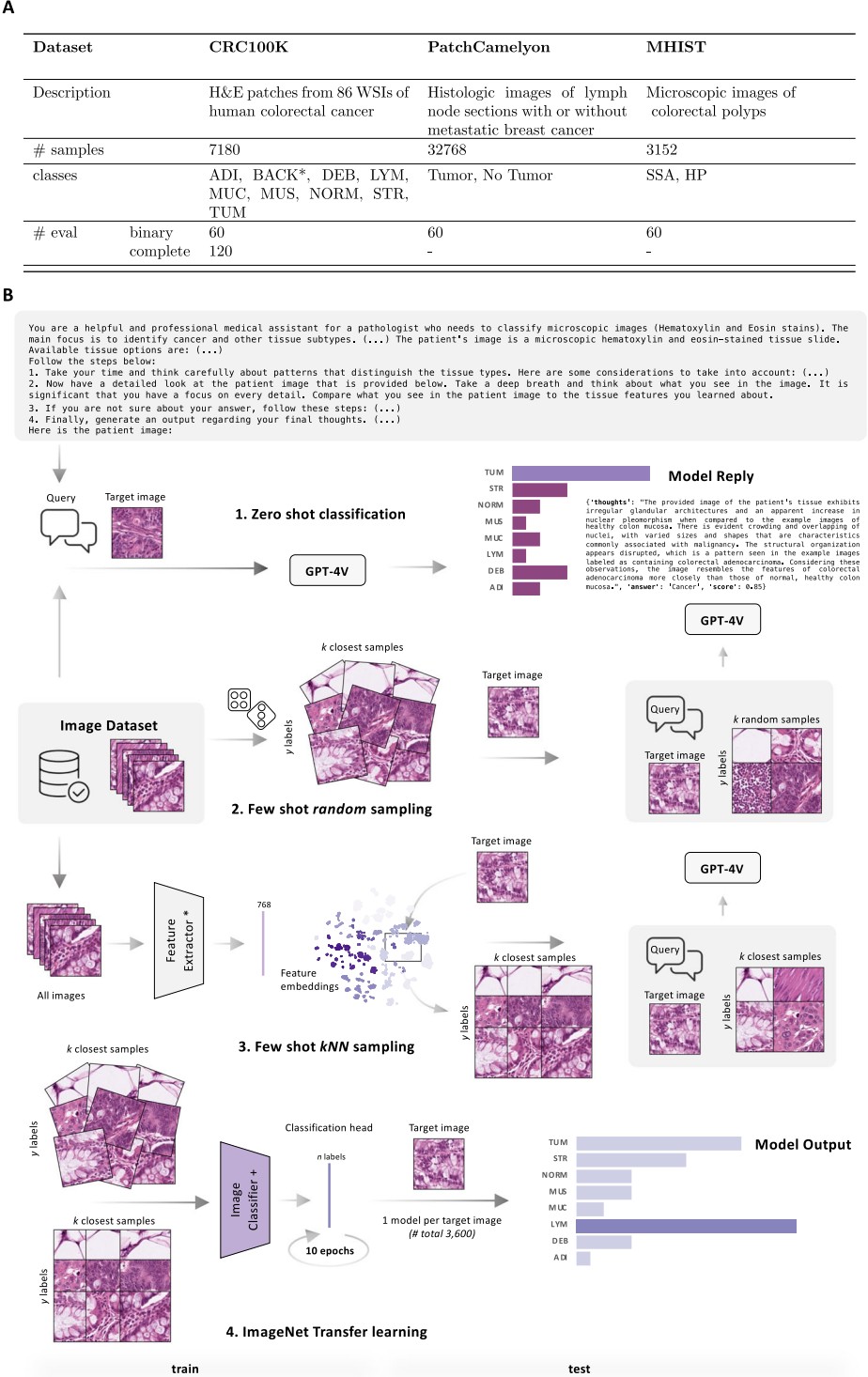

**Fig. 1 | Comprehensive schematic.** This figure presents a systematic overview of the three histopathology benchmarking datasets, detailing the number of samples incorporated in our study (Panel **A**). A selection of random test images was drawn from each of these datasets for evaluation using three distinct methodologies: Zero-Shot Classification (Method 1), random few-shot sampling (Method 2), and *kNN*-based selection (Method 3). For the latter, feature extraction was performed using the *Phikon* ViT-B 40 M Pancancer model (*). Cosine similarity was used as the comparison metric between the target image and its closest *k* neighbors in embedding space. As a benchmark against GPT-4 *ICL*, we trained four image classifiers (indicated by +, namely ResNet-18, ResNet15, Tiny-Vit, and Small-Vit) via transfer learning from ImageNet for each target image (Panel **B**). For an in-depth understanding of these methods, please refer to Algorithm 1 and the Experimental Design section. * The BACK (background) label was excluded from the analysis.

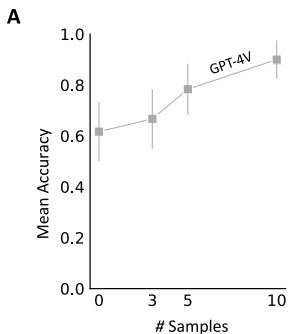
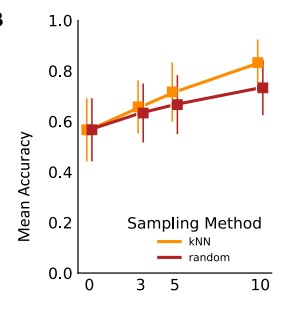
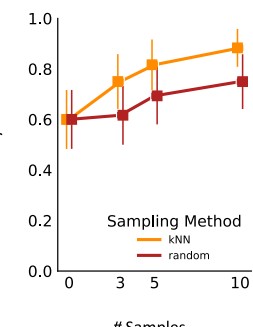

**Fig. 2 | In-context learning for vision-language models.** Panel **A** shows that classification accuracy on a simple task detecting tumor (TUM) versus non-tumor (NORM) tiles from the CRC100K dataset can drastically be improved by leveraging ICL through randomly sampled, few-shot image samples. Additionally, we compare random and kNN-based image sampling on two datasets and show that *kNN*-based image sampling improves model performance in classifying images from both MHIST (left) and PatchCamelyon (right), especially when scaling the number of few-shot samples (Panel **B**). Note that samples have been slightly shifted on the x-axis for visibility. The *y*-axis denotes the mean accuracy with lower and upper 2.5% confidence intervals (CIs) from 100,000 bootstrap iterations for both panels, respectively. Source data are provided as a Source Data file.

potential application, which limits researchers to develop these models at scale.

In-context learning (*ICL*)—a concept borrowed from the field of natural language processing (NLP)—could provide a possible solution to this problem. The ability of large language models (LLMs) to learn from a few handcrafted examples that are provided to the LLM alongside the prompt, holds great potential and has been shown to improve model performance[12]. A practical implementation in a medical setting might involve presenting the LLM with a detailed clinical scenario, such as a complex oncology case, accompanied by several comparable instances with different strategies on how to solve a certain challenge. This approach is called few-shot prompting. Numerous methodologies have been developed utilizing in-context learning. Their foundational principles are explained in detail in the 'Supplementary Methods: In-Context Learning' section.

In the medical field, one model has recently been built upon the aforementioned paradigms: MedPrompt[13], which is based on the GPT-4 architecture. Central to this method is the application of the k-nearest Neighbor (*kNN*) search, which herein helps identify the most relevant few-shot examples for a specific clinical input. This process involves comparing text embeddings, which are numeric representations of words with the input in question, and then selecting samples with the closest alignment. We highlight further implementation details of this approach, as it has partial overlap with the methods developed in our study, in the 'Supplementary Methods: Related Work −Enhancing LLM strategies' section.

However, a major shortcoming is the restriction to text-based tasks. Medicine is a highly multimodal discipline, where a comprehensive understanding of a patient's symptoms or diagnoses requires information from diverse data sources such as radiographic and microscopic imaging, clinical reports, laboratory values, and electronic health records[14]. Only recently, the AI community has entered into the field of vision language models (VLMs), exemplified by the release of GPT-4V[15], the announcement of Google DeepMind's Gemini[16] family or open-source variants like LLaVA[17].

Building on the trend of large vision language foundation models, we hypothesize that the principles applied for in-context learning of text-based models can be equally effective when extended to multimodal scenarios, such as medical imaging. In the non-medical setting, robust evidence for in-context learning with images has already been established[18]. Especially, in the medical field, where generating annotated ground truth data presents a critical challenge, the potential for performance improvements through this approach could be immensely beneficial. This issue is also of relevance for underrepresented medical cases, such as rare tumor types, which receive insufficient

representation in traditional deep-learning training pipelines. Moreover, the concurrent integration of textual, theoretical knowledge, and visual information could pave the way toward a more holistic understanding of multidimensional medical data.

In this study, we present results of benchmarking the efficacy of in-context learning with GPT-4V against dedicated image classifiers across three histopathology benchmarking datasets. Notably, we demonstrate that the performance of GPT-4V in tissue classification can be improved through in-context learning and is on par with specialist computer vision models. This advancement casts doubt on the necessity of developing task-specific deep learning models in the future and democratizes access to generalist AI models to accelerate medical research.

## Results

### In-context learning with medical images improves classification accuracy for histopathology

In this study, we hypothesize that few-shot prompting can improve the performance of foundation vision models. This hypothesis has been shown with text-only tasks, but remains unclear for its application to biomedical images[12,18]. We provide a high-level overview of our evaluation datasets (Fig. 1A) and the overall experimental concept in Fig. 1B. We first evaluate this hypothesis on a binary classification task between tumor (TUM) and non-tumorous normal mucosa (NORM) tissue tiles from the CRC100K dataset[19]. As shown in Fig. 2A, GPT-4V only marginally surpasses the expectation of random guessing when used in a zero-shot setting, attaining an accuracy of 61.7% (CI: 0.5–0.733). In-context learning changes this situation: We see a consistent improvement in classification accuracy with increasing numbers of few-shot samples with an accuracy of 66.7% in the three-shot sampling setting (CI: 0.55–0.783), 78.3% for five-shot sampling (CI: 0.667–0.883) and an accuracy of 90% when showing 10 images of each class to the model (CI: 0.817–0.967). In our subsequent ablation study (Fig. 2B), we compare random versus kNN sampling across the MHIST[20] and PatchCamelyon[21] (PCAM) datasets. From a zero-shot baseline that again barely achieves a better classification than random guessing (MHIST accuracy 56,7%, CI: 0.433–0.683; PCAM accuracy 60%, CI: 0.467–0.717), we see that in both datasets, random image sampling can improve classification accuracy. These results can further be improved by selecting the sampled images based on their similarity to the target image (kNN sampling), which results in the best-achieved accuracy of 83.4% and 88.3% for detecting sessile-serrated adenoma over hyperplastic polyps (MHIST, CI: 0.733–0.917) and lymph-node metastases from breast cancer versus tumor-free lymphatic tissue (PCAM, CI: 0.8–0.95) in a ten-shot setting.

In summary, these results demonstrate that in-context learning can improve the performance of foundation vision models in classifying histopathology images. Moreover, we show that *kNN* sampling can further enhance accuracy over random sampling, especially when increasing the number of images that are shown to the model. Corresponding metrics can be found in Tables 1 and 2, with the best-performing method highlighted in bold.

### Vision-language models can achieve performance on par with retrained vision classifiers

Next, we compare few-shot sampling with the previous status-quo[7] in image classification, which involves retraining models from ImageNet weights. As an initial comparison, we train one distinct model for each target image shown to GPT-4V, with the identical images used for in-context learning as the training set. This approach reveals that in-context learning is sufficiently robust to achieve results that are on par with, or even surpass, specialized narrow image classifiers under the same conditions. Specifically, the ten-shot in-context learning GPT-4V approach not only matches but exceeds the performance of all other models (Fig. 3A), leading to a classification accuracy of 83.3% for MHIST (CI: 0.733–0.917) and 88.3% for PatchCamelyon (CI: 0.8–0.95), outperforming the second-best model, Tiny-ViT, by 3.3% and 6.6% respectively. Notably, in the case of PatchCamelyon, even the three- and five-shot prompting were sufficient to outperform all other models in this setting. We show a detailed comparison of each model's evaluation metrics in Supplementary Table 1. Further, we extend this comparison to include two scenarios: Firstly, we show that GPT-4V's performance through in-context learning can partially match that of previously mentioned vision models, even when those models have been trained on the complete training datasets, such as CRC100K which includes tens of thousands of tiles (Supplementary Table 2). Secondly, we compare the efficacy of in-context learning with the current gold standards in histopathology image classification—specifically using the Phikon and UNI models as examples. All evaluation metrics, including specifications on training parameters, are shown in Supplementary Tables 3 and 4. The results indicate that in-context learning significantly narrows the performance gap between GPT-4V and these models. For instance, in-context learning reduced the disparity from 36.6% (zero-shot GPT-4V) to a 10% difference (ten-shot GPT-4V) relative to the performance achieved by *kNN* classification with the Phikon feature extractor on the MHIST dataset. We also discovered that GPT-4V demonstrated remarkable zero-shot capabilities for some of the targets: For PatchCamelyon, it correctly identified all tumor tiles, albeit with a high false positive rate of 80%. In the MHIST dataset it correctly recognized 83% of Sessile Serrated Adenomas but

only 30% of Hyperplastic Polyps (Fig. 3B). Considerable improvements could be observed with few-shot prompting. In the case of PatchCamelyon, the model's ability to identify normal lymph node tissue progressively increased with the number of example images, ranging from an accuracy of 67% for three-shot, 77% for five-shot to 80% for ten-shot image prompting. Similarly, for MHIST, the correct identification of hyperplastic polyps could be increased from 30% (zero-shot) to close to 90% (ten-shot). Notably, these enhancements did not compromise the model's performance in detecting tumors in the PatchCamelyon dataset or SSAs in the MHIST dataset (Fig. 3C). These findings show that in-context learning with microscopic images can achieve an accuracy on par with fine-tuning specialized image classification models.

### In-context learning reduces the performance gap between generalist and histopathology foundation models

In a subsequent evaluation, we tested GPT-4V on the CRC100K dataset, which is more challenging as it consists of a more diverse set of labels. As we increased the number of few-shot image samples, GPT-4V showed considerable improvements in performance. However, it did not achieve the levels observed on other datasets such as PatchCamelyon or MHIST (Fig. 4A). Despite this, there was a significant narrowing of the performance gap between GPT-4V and models fully trained on all data, such as ResNet-15, ResNet-18, ViT-Tiny, and ViT-Small, as well as compared to the downstream performance of the Phikon and UNI models. Initially, the performance deficit of GPT-4V in zero-shot classification relative to *kNN* stood at 61.7% and 62.5% for Phikon and UNI respectively. This gap was reduced to 18.3% and 19.2% when using five-shot in-context learning compared to the best scores achieved by Phikon and UNI in any of our settings. While our study does not claim to maximize potential performance across all models, it highlights that ICL can bring us closer to the performance levels of models extensively pretrained on these tasks. Also, the GPT-4V model natively excelled in identifying tumor and muscle tissue, achieving a recall score of 80% and 100%, respectively. However, it failed completely in recognizing debris (DEB), adipose tissue (ADI), lymphocytes (LYM), mucus (MUC), and tumor-associated stroma (STR). Herein, three instances are particularly noteworthy: lymphocytes were consistently misclassified as tumor tissue, debris was incorrectly categorized as a tumor in 93% of cases, and stroma was misclassified as muscle tissue in 87% of instances. The addition of few-shot examples led to a substantial improvement. The best results are achieved with five-shot *kNN*-sampling, where the model receives a total of 40 sample images. This leads to enhanced accuracy across all labels (Fig. 4B). A clear trend of continuous performance gains is evident as the number of few-shot samples is increased, demonstrating consistent improvements at each stage of the process (from zero- to one-, one- to three-, and three- to five-shot prompting) for almost all labels (LYM, MUC, NORM, STR), with the exception of debris (Fig. 4C). Details to confidence intervals are summarized in Supplementary Table 1. In summary, our findings underline the potential of few-shot image learning in GPT-4V, even in a multilabel classification setting.

### Table 1 | Metrics for GPT-4V few-shot sampling

| Model | Metric | Zero | Three | Five | Ten |
|---|---|---|---|---|---|
| GPT-4V | Accuracy | 0.617 | 0.667 | 0.783 | **0.9** |
| | Lower CI | 0.5 | 0.55 | 0.667 | 0.817 |
| | Upper CI | 0.733 | 0.783 | 0.883 | 0.967 |

### Table 2 | Metrics for random and *kNN* few-shot learning with GPT-4V on MHIST and PCAM

| Sampling | Metric | MHIST | | | | PCAM | | | |
|---|---|---|---|---|---|---|---|---|---|
| | | Zero | Three | Five | Ten | Zero | Three | Five | Ten |
| Random | Accuracy | **0.567** | 0.633 | 0.667 | 0.733 | **0.6** | 0.617 | 0.694 | 0.75 |
| | Lower CI | 0.433 | 0.517 | 0.55 | 0.617 | 0.467 | 0.5 | 0.581 | 0.633 |
| | Upper CI | 0.683 | 0.75 | 0.783 | 0.833 | 0.717 | 0.733 | 0.806 | 0.85 |
| *kNN* | Accuracy | **0.567** | **0.658** | **0.717** | **0.834** | **0.6** | **0.75** | **0.817** | **0.883** |
| | Lower CI | 0.45 | 0.553 | 0.6 | 0.733 | 0.467 | 0.633 | 0.717 | 0.8 |
| | Upper CI | 0.683 | 0.763 | 0.833 | 0.917 | 0.717 | 0.85 | 0.9 | 0.95 |

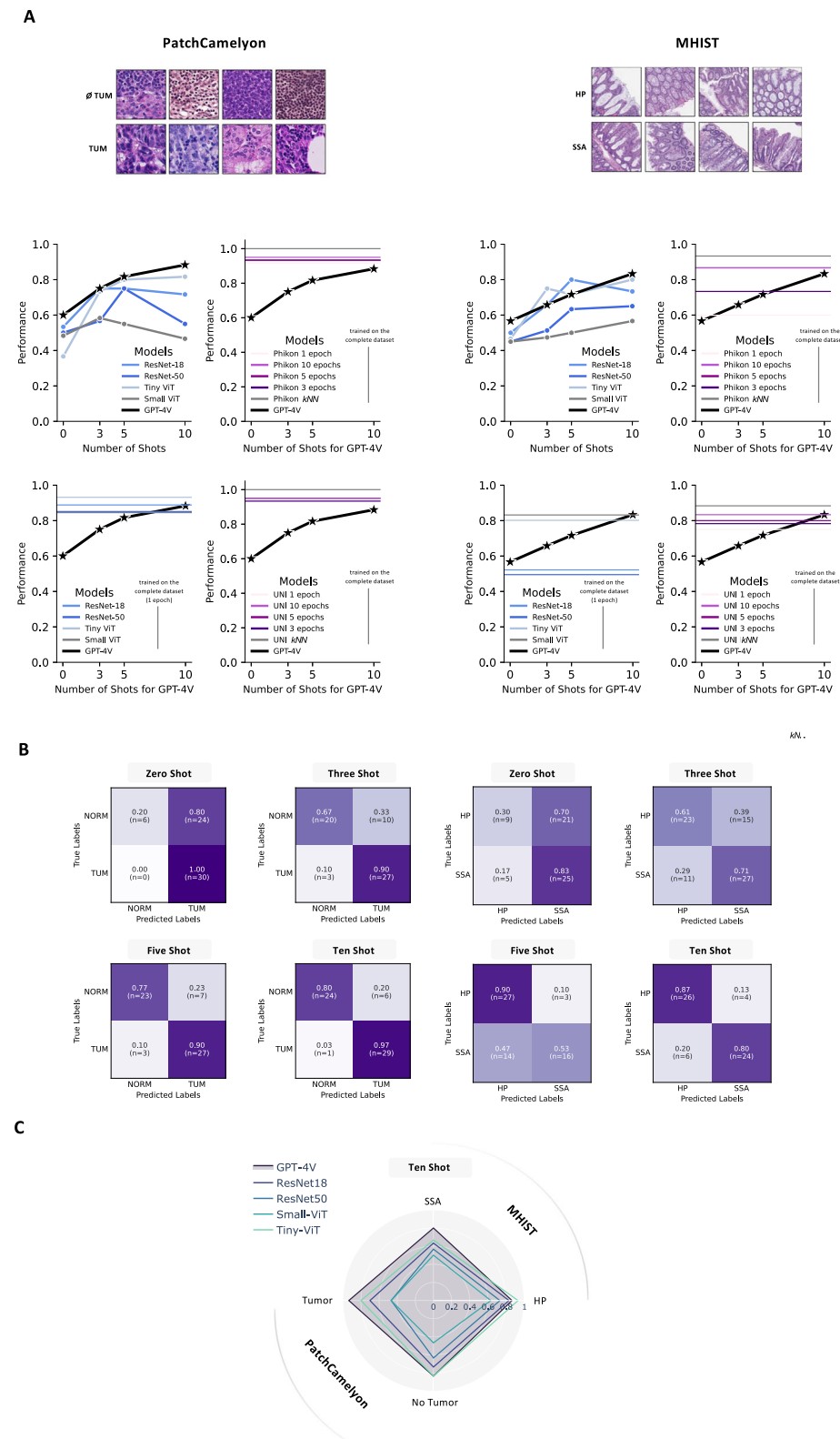

## Image in-context learning improves text-based reasoning

Vision-Language Models enable multimodal understanding. To more accurately evaluate the impact of few-shot image sampling on textual reasoning within VLMs, we further investigated the output of GPT-4V and created text embeddings using Ada-002. Next, we utilized t-Stochastic Neighbor Embedding (t-SNE) to analyze the semantic space of the model's reasoning. Our results demonstrated distinct clusters of embeddings when compared to the model's final answer (Fig. 5A, top), underscoring a potential correlation between text and image data. However, in the zero-shot scenario, the comparison of text embeddings to ground truth labels revealed that the model's intrinsic reasoning only correlated poorly with the correct categorization of the images (labels). This suggests a limitation in the model's ability to independently navigate to the correct label based solely on its learned

**Fig. 3 | Performance analysis of GPT-4V with *kNN* ICL on PatchCamelyon and MHIST datasets.** This figure is divided into two sections, with Panel **A** and **B** focusing on PatchCamelyon (to the left) and the MHIST dataset (right subpanel) respectively. In **A**, line graphs illustrate the average performance of GPT-4V when used with *kNN*-based in-context learning relative to several specialist image classification and histopathology foundation models: We first compare GPT-4V with ResNet-18, ResNet-50 and two Vision Transformers (ViT-Tiny and ViT-Small) where the number of ICL samples for GPT-4V equals the number of training samples for the image classification models (1, top left). Additionally, we compare the same vision classifiers, trained on the full respective datasets (2, bottom left), and the performance of two histopathology foundation models, Phikon (3, top right) and UNI (4, bottom right). For the latter, we compare GPT-4V against training a linear layer on top of the pre-trained foundation model (for one, three, five, and ten epochs) and *kNN* classification. Note that in these cases, the models are trained on the full datasets, and the term '# Samples' is used to denote the number of few-shot ICL samples for GPT-4V only. The Y-axis displays the average accuracy across all labels, derived from 100,000 bootstrapping steps. All relevant metrics (accuracy, lower and upper confidence intervals) are summarized in Supplementary Tables 1–3. Panel **B** presents a series of heatmaps, highlighting the absolute and relative performance per label in zero-, three-, five-, and ten-shot *kNN*-based sampling scenarios, each with a sample size of $n = 60$. Lastly, the spider plot in Panel **C** highlights the superiority of 10-shot GPT-4V in classification performance for both datasets when compared under equitable conditions to two ResNet-style models and two vision transformers. Source data are provided as a Source Data file.

representations. Contrastingly, the application of few-shot learning techniques improved the separation of text embeddings corresponding to different answers and labels. This enhancement is evident from the formation of a greater number of distinct clusters and more accurate alignment of data points with their respective ground truth categories, as shown in Fig. 5A. Moreover, the implementation of few-shot learning was associated with increased silhouette scores, indicating closer proximity of data points to their correct labels as the number of example images provided to the model increased. Collectively, these findings suggest that employing few-shot learning techniques can enhance the model's capacity to analyze and interpret test images more accurately, thereby refining its decision-making process.

To showcase the benefits multimodality might have in histopathology, we present two illustrative cases from our study. Figure 5B (left) depicts a scenario where GPT-4V falsely classifies an image as a tumor, while the underlying ground truth was considered to be stroma.

However, GPT-4Vs detailed reasoning, identifying morphological signs indicative of cancer, reveals the presence of tumor cells characterized by irregularly shaped nuclei. Analyzing the 500 closest image embeddings in feature space shows a similar trend, with two-thirds of image embeddings being categorized as tumors. Another case, shown in Fig. 5B (right), demonstrates GPT-4V's proficiency in transferring knowledge from different domains to draw the right conclusions. According to existing literature[22], the term "chicken wire pattern" is established within the domain of pathology, yet only regarding the appearance of adipose tissue in liposarcomas and other malignancies. However, it is not frequently used to describe the architecture of normal, healthy adipose tissue. The capability of GPT-4 to transfer its understanding of the physical appearance of chicken wire to the shape of adipose tissue in histopathology demonstrates the ability for transfer learning and holds potential in areas like AI explainability and teaching. Overall, these data indicate that vision language models possess substantial potential for medical image classification in histopathology, utilizing only a few sample images. This capability may provide inherent advantages over traditional image classifiers, due to their multimodal architecture.

## Discussion

Foundation models have demonstrated substantial promise in medical image processing. Zhou et al. trained such a system using 1.6 million retinal images and illustrated that they could then fine-tune it with fewer annotated images to assist clinicians in identifying a range of ocular diseases[23]. Yet, the vast amount of data that is required and the necessity to develop one specific fine-tuned version for each clinical task, currently constrain training these models at scale, limiting their utility to researchers with extensive knowledge in computer sciences and access to the required hardware. Furthermore, the applicability of these models has been confined to the visual field only. Nonetheless, learning is a multimodal process. For example, in pathology, practitioners and students assimilate their knowledge by extracting visual patterns from images and synthesizing them with corresponding

textual annotations. In summation, the ideal scenario would envision AI systems that seamlessly combine multimodal information in a data-efficient manner while having the flexibility to adapt their behavior to any given task on demand without the need for traditional retraining.

In this study, we demonstrate a proof of concept illustrating that achieving these properties is possible with in-context learning on vision language models, exemplified on GPT-4V: We show that this method not only is effective when classifying medical microscopy images but also that it can achieve performance comparable to conventional image classification models and that in-context learning provides a data- and resource-efficient learning method to drastically recover the performance gap between generalist foundation models and histopathology foundation models like Phikon and UNI, that are trained on a large corpus of microscopic images. We show that five to ten sample images per label are enough for GPT-4V to achieve classification accuracy scores close to the current gold standard models. These results are encouraging, especially considering that other current state-of-the-art pathology foundation models like Paige's Virchow[24] report performance metrics that marginally surpass our method, with reported accuracy scores of 82.7% compared to 83.3% for GPT-4V on the MHIST dataset and 92.7% versus 88.3% for GPT-4V on PatchCamelyon. For MHIST, we must note here that we excluded images without a full inter-rater agreement, which makes our use case most likely easier than the one used by Vorontsov et al.[24]. We acknowledge the lack of public access to the training corpus of GPT-4V, which raises the possibility that the model may have been trained on our test sets. Nevertheless, the performance observed in a zero-shot scenario marginally surpasses random guessing, making it less likely that the data had been used for training. We use this zero-shot baseline as a comparison to investigate the benefit of in-context learning. With our approach, we lay the foundation for a general-purpose framework that advances state-of-the-art prompting techniques for images. Additionally, our results indicate that deliberately selecting few-shot examples that are semantically similar to the test image can substantially improve the performance of the model. A notable aspect is the integration of text with vision, which can help in explainability in understanding a model's reasoning processes. This addresses a critical limitation of conventional image classifiers, as textual feedback provides a more comprehensible way of understanding and interpretability for humans compared to visual tools such as Grad-CAM[25]. This aspect is crucial for reliable AI systems in medical applications[26].

Some limitations of our work are that experiments were restricted to a yet small sample size due to the preview status of the GPT-4V API, which currently only permits a limited number of requests. Another limitation in this regard is that we did not include *ensembling* methods, which would require multiple model iterations over the same task as performed by MedPrompt or Med-PaLM 2, as this approach has a total of 44 model calls for a single task only[27]. Moreover, it is worth noting that the performance of in-context learning with images sometimes yields suboptimal results, particularly in classes like debris, mucus, and stroma within the CRC100K dataset. This observation is in line with

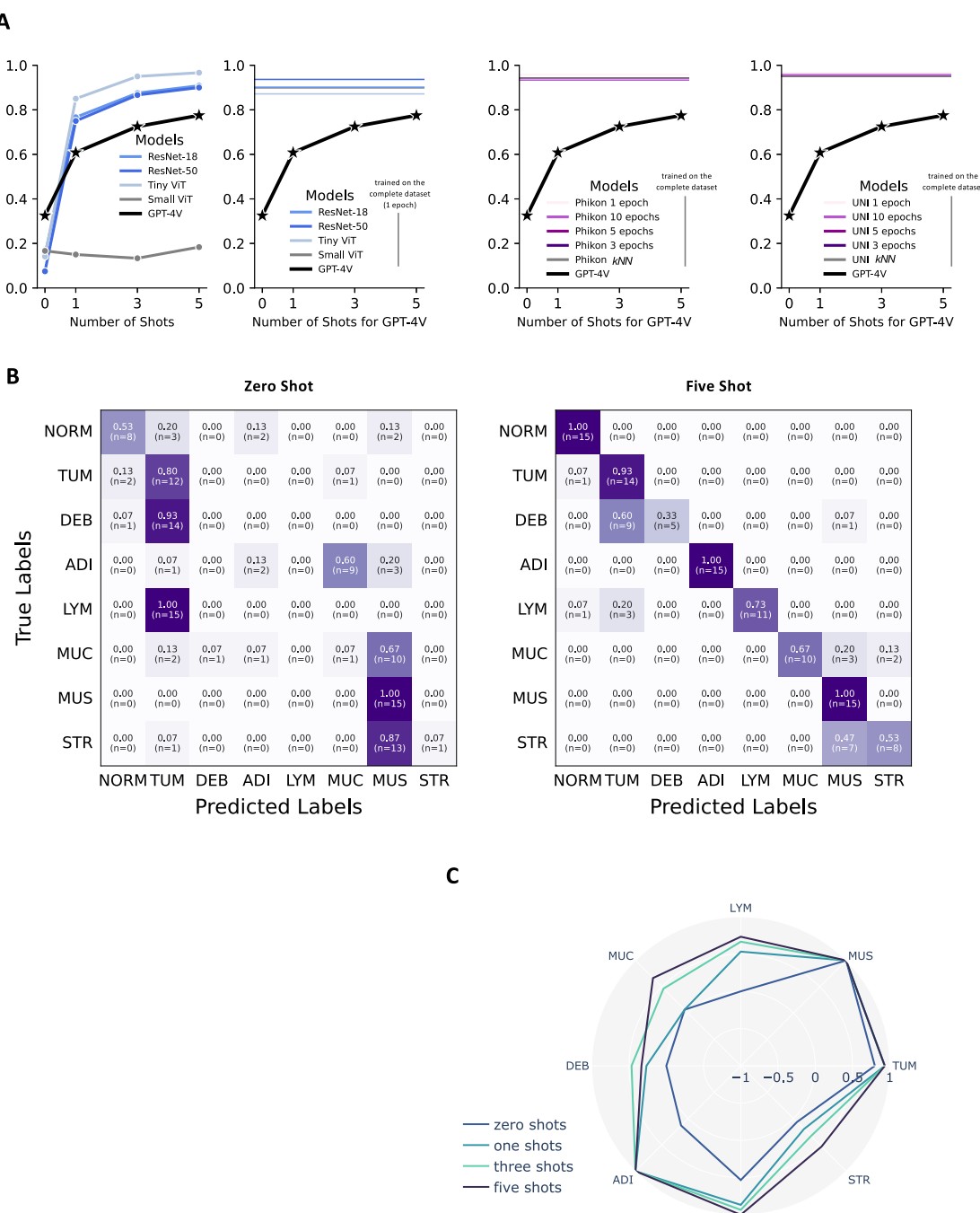

**Fig. 4 | Performance analysis of GPT-4V with *kNN*-based sampling on the CRC100K dataset.** The line graphs (Panel **A**) show the comparative average performance of GPT-4V with kNN-based in-context learning against the four image classification models (1) when trained on the same number of images as used as example images for in-context learning with GPT-4V. Additionally, we show how in-context learning can reduce the performance gap between GPT-4V and the respective image classifiers when trained on the entire datasets respectively (2) as well as in comparison to the state-of-the-art foundation models Phikon and UNI. # Samples refers to the count of few-shot *ICL* samples for GPT-4V and training samples for the other models in 1, while for all other settings, the models are trained on the entire training data. The *y*-axis represents the mean accuracy across all labels, computed using 100,000 bootstrapping iterations. Detailed average accuracy values, including confidence intervals, are summarized in Supplementary Table 1. Panel **B** features confusion matrices for GPT-4V in both zero-, and five-shot *kNN*-based sampling scenarios (*n* = 120 samples). The spider plot showcases the average classification accuracy per label per number of *kNN*-sampled shots, revealing a general trend towards increased classification accuracy across most labels with scaling of the number of few-shot image samples (Panel **C**). Source data are provided as a Source Data file.

findings by Huang et al. [28]. While these outcomes have been acknowledged, we leave an in-depth investigation into the underlying reasons and the development of potential solutions as subjects for future research. Moreover, further explorations into prompt engineering techniques beyond Chain-of-Thought, like Tree-of-Thought[29] could be

used to optimize the models beyond the current results. Finally, the current reliance of our approach on generating and retrieving image embeddings through a specialized vision model (i.e., *Phikon*) is a drawback to the philosophy of an all-encompassing foundation VLM, which ideally would not require a specialized vision encoder to

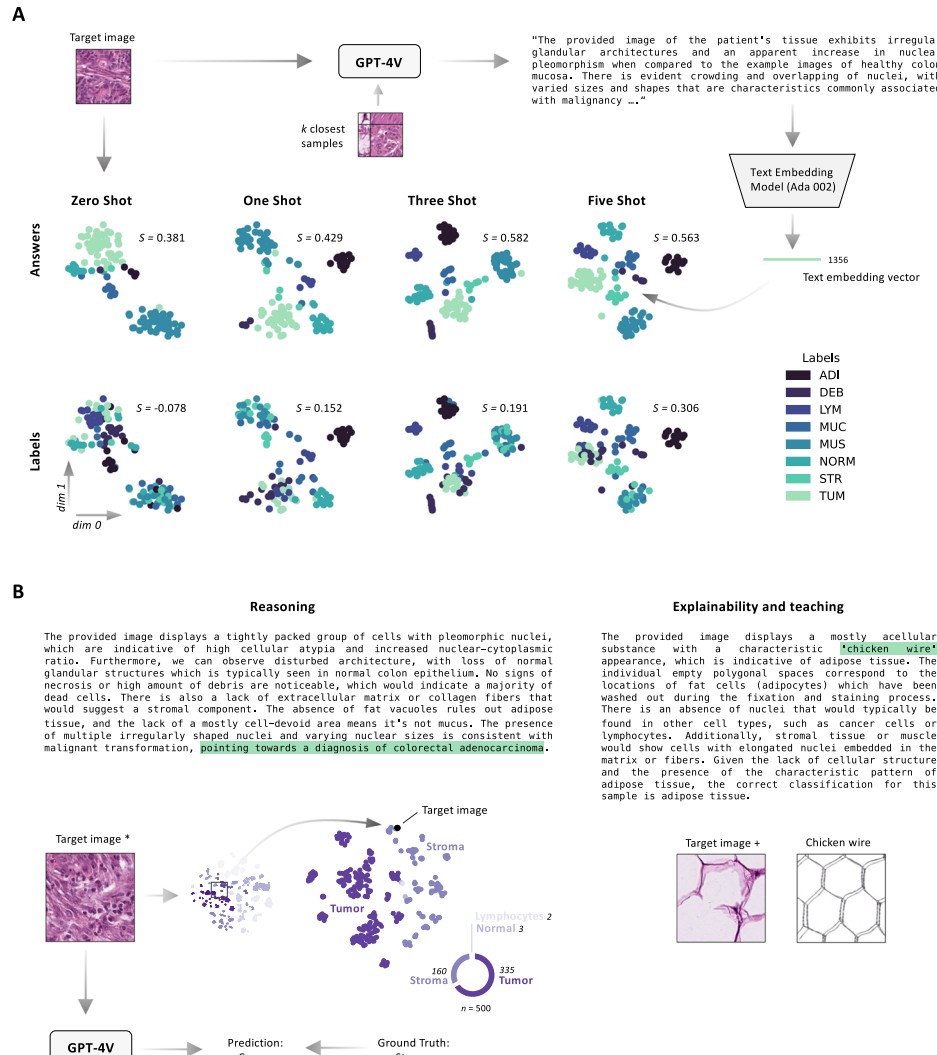

**Fig. 5 | Few-shot sampling improves text-based reasoning.** Panel **A** depicts the workflow, starting from GPT-4V's initial prediction and its reasoning process ('thoughts'), to the generation of text feature embeddings with Ada 002. The panel of t-SNEs demonstrates the evolution from a zero-shot framework on the far left, advancing through one-, three-, and five-shot *kNN* sampling to the right. All data is obtained from the CRC100K dataset. In the t-SNE plots, color coding distinguishes between the model's final classifications ('Answers', top) and the ground truth ('Labels', bottom). The introduction of few-shot image sampling noticeably refines the model's textual reasoning, as evidenced by the formation of more distinct clusters in alignment with the model's own responses (top) and the underlying ground truth (bottom). *S* denotes silhouette scores, which are calculated for each t-SNE. Complementary to these visualizations, Supplementary Fig. 2 features word clouds that further illustrate the alignment of the model's vocabulary with clinical diagnoses, highlighting key terms such as "lymph node" for normal tissue and "metastatic / breast cancer" for malignancies, thereby enhancing the interpretability of the model's diagnostic reasoning process. In Panel **B**, we present two exemplary scenarios to demonstrate the potential superiority of integrated vision-language models over stand-alone image classification models. On the left, an image is displayed where the original annotation identified the sample as stroma (STR), yet GPT-4V categorizes it as tumor (TUM). The rationale provided by the model appears plausible, notably pointing out several abnormally shaped nuclei, visible, for instance, in the lower right corner. This sample indeed appears to represent a borderline case. When comparing the top 500 closest patch embeddings to the reference image, a dominant fraction is classified as tumor (67%), with a lesser proportion being labeled as stroma (32%) and a negligible percentage (<1%) as lymphocytes or regular colon epithelium. The exploration of GPT-4V's interpretive process can help identify and understand such complex edge cases that go beyond what is possible with conventional image classifiers alone. Right: Chicken-wire patterns are described in the histology of liposarcoma, which arises from adipocyte precursor cells. This description stems from its resemblance to chicken wire fences (shown to the right). GPT-4V effectively leverages this knowledge from another context to describe the morphology of the adipocytes shown in this image. This way of performing 'transfer learning' could have strong implications in teaching. * The image name in the CRC100K cohort is STR-TCGA-VEMARASN. + The image name in the CRC100K cohort is ADI-TCGA-QFVSMHDD.

generate features. Although we have not tested this at scale due to the rate limits of the API, we speculate that the next generation of foundation models will be able to autonomously manage, embed, and retrieve sample data on demand for few-shot learning. Following the current paradigm of AI scaling laws[30,31], it can be estimated that we have not yet reached a plateau in the performance benefits from even more powerful foundation models in the future. Furthermore, our experiments have not indicated any saturation point in model efficacy when increasing the number of *k*-shot examples—however further

increasing the number of sample images per task was not feasible due to exceeding the models context window. Again this suggests the potential for continued enhancements when further scaling our approach and raises the question about the necessity and efficacy of researchers to develop their own specialized deep learning models for each task, particularly when a singular model may suffice in the foreseeable future. In summary, we further aim to scale our work to overcome these limitations and extend it to other domains like radiology imaging. Nevertheless, we believe that in-context learning with

images holds great potential for improving the performance of vision language models on biomedical image classification tasks and beyond.

## Methods

### Ethics statement

This study does not include confidential information. All research procedures were conducted exclusively on publicly accessible, anonymized patient data and in accordance with the Declaration of Helsinki, maintaining all relevant ethical standards. The overall analysis was approved by the Ethics Commission of the Medical Faculty of the Technical University Dresden (BO-EK-444102022).

### Datasets

Our benchmarking experiments are conducted on the following, open-source histopathology image datasets:

- **CRC-VAL-HE-7K**[19] is the evaluation set associated with the NCT-CRC-HE-100K dataset, consisting of 7180 image patches extracted from hematoxylin & eosin (H&E) stained formalin-fixed and paraffin-embedded (FFPE) sections from 50 individuals with colorectal cancer. Samples were collected at the NCT Biobank (National Center for Tumor Diseases, Heidelberg, Germany) and the UMM pathology archive (University Medical Center Mannheim, Mannheim, Germany) and digitized at 224 × 224 pixels (px) at a resolution of 0.5 microns per pixel (MPP). Throughout this manuscript, we will refer to this dataset as *CRC100K*. Following previous studies[9,32], the background (BACK) class was excluded from our analysis.
- **PatchCamelyon (PCam)**[21] contains 327,680 H&E stained histologic image patches at 96 × 96px (0.243 MPP) from human sentinel lymph node sections obtained from the Camelyon16 Challenge, originally split into a training and validation set. Samples are annotated with a binary label to denote the presence or absence of metastatic breast cancer tissue at a balance close to 50/50.
- **MHIST**[20] is a dataset of 3152 H&E-stained FFPE-sections from colorectal polyps, collected at the Dartmouth–Hitchcock Medical Center (DHMC) and addresses the challenging problem of discriminating sessile serrated adenoma (SSA) from hyperplastic polyps (HP)[33]. Images are scanned at 224 × 224 px and labeled as either HP or SSA by the majority vote of seven pathologists, resulting in a 3:7 split.

For GPT-4V inference testing, we randomly generated randomly chosen test datasets containing 60 samples for MHIST and PatchCamelyon and 120 samples for CRC100K at a balanced 1:1 split for each of the available labels. For simplicity, we restricted the test images from the MHIST dataset to those achieving unanimous expert consensus for the presence of SSA or HP, respectively. All images that were used for inference testing are visualized in Supplementary Fig. 1.

### GPT-4V model specifications

All experiments in this study were performed using the GPT-4V model in the chat completions endpoint of the official OpenAI Python API between November 15 and December 03, 2023. The official model name in the OpenAI API is *gpt-4-vision-preview*. For simplicity, we will use the term GPT-4V in all subsequent references to this model throughout our manuscript. Temperature was set to 0.1 based on initial experiments and no other modifications to model hyperparameters were made. For further implementation details, we refer to our official github repository.

Text embeddings were created using OpenAI's default embedding model Ada 002, without further modifications.

### Prompting and random few-shot image in-context learning

In the following, we present a brief overview of the implementation of the final prompts used in GPT-4V. For an in-depth explanation of both the system prompt (instructions dictating the expected model behavior) and user prompt (input commands or queries to the model), please refer to Supplementary Tables 5–7. There is currently no standardized blueprint for the development of effective model prompts; rather this is an iterative, dynamic process driven by trial and error. Our prompting strategies were developed on a selection of ten random image tiles per label from each dataset. Following current best practices, we utilized the system prompt to establish the setting (context) of the model and to guide its expected behavior. In our initial trials with GPT-4V, we encountered several limitations due to the model's intensive policy alignment regarding its refusal to handle medical data. To address these issues, we modified our approach by presenting test cases as hypothetical scenarios (*'None of your answers are applied in a real-world scenario or have influences on real patients.'*) and additionally included a selection of desired and undesired response pairs into the system prompt. To simplify the analysis of the results, we also configured GPT-4V to generate answers in JavaScript Object Notation (*JSON*) format. This included a structured template containing a field for providing logical reasoning (*'thoughts'*), the final *'answer'* as well as a certainty *'score'*.

Regarding the user prompt, we differentiate between the zero- and few-shot settings. In the zero-shot scenario, we started with enumerating all possible label options, followed by guiding the model to adopt a step-wise reasoning akin to Chain-of-Thought (*CoT*) prompting[29]. This was followed by a compilation of dataset-specific considerations: For instance, in the *CRC100K* dataset, we observed that the model would almost always choose to classify an image tile as a tumor whenever detecting malignant cells, despite simultaneously recognizing the major cell fraction being lymphocytes. To counteract these dataset-specific pitfalls, we included concise guidelines at this step (Supplementary Tables 5–7). Finally, GPT-4V was asked to thoroughly examine the appended patient image and provide its answer as described above.

In the few-shot sampling prompts, we presented a sequence of $k$ example images (where $k$ equals 1, 3, 5, or 10), each followed by its corresponding label, in a repeated pattern: Specifically, we presented a single image corresponding to each label $y$, cycling through the entire set of labels $k$ times. This means that for every test image, we present the model a total of $(k * y) + 1$ images. This setting is the same for the *kNN*-based sample selection, which we further highlight below and in Box 1. Each image was prefaced with the phrase '*The following image contains {y}:*' for any possible label $y$. Then GPT-4 was instructed to closely compare and extract meaningful knowledge from the images for subsequent comparison with the target image. Beyond this, the structure of the prompt remained consistent with the zero-shot template. Moreover, to the best of our knowledge, we followed all known best practices and prompting tricks (i.e., *'Take a deep breath'*)[34]. To mitigate the risk of overfitting the samples used during the refinements of the system and user prompts, we ensured that they were not included in the generation of inference test data. More specifically, we performed an initial investigation of zero-shot and random few-shot performance using an initial dataset comprising 30 random samples, collected exclusively from the CRC100K dataset, each containing either tumor or normal colon epithelium. This initial dataset served a dual purpose: developing effective prompts and providing an early insight into model responses. For the following evaluation phase, we collected a new subset of 30 samples. This way, we prevented sample leakage from our prompt creation dataset into our final evaluation test set. This was critical to prevent overfitting that could arise from sample-specific biases we might have included in the prompt. However, these samples were allowed to be part of either random or (as described in the next section) *kNN*-based sample selections. This process was repeated for every dataset.

## BOX 1

# Pseudocode for the entire KNN-sampling process

**Algorithm 1** $k$-NN few-shot image sampling

**Require:** Number of closest neighbors $k$.

**Require:** Encoder function Encoder($\cdot$) that maps an image to an embedding.

**Require:** Language model LLM($\cdot$).

**Require:** System prompt $P_{sys}$ that describes the expected model's behavior.

**Require:** Dataset $\mathcal{D} = \{(x_i, y_i)\}_{i=1}^{N}$ containing $N$ image-label pairs, each with an image $x_i$ and a label $y_i \in \mathcal{Y}$ ($\mathcal{Y}$ is the set of all possible labels).

**Require:** Task list $\mathcal{T} = \{(P_t, x_t, y_t)\}_{t=1}^{T}$ containing $T$ tuples of user prompt $P_t$, target image $x_t$ and ground truth label $y_t$.

**Ensure:** Final result $\mathcal{R}$ for each task in JSON format as {thoughts, answer, score}.

1: Initialize an empty mapping $E$ from image ID to embedding.
2: **for each** image-label pair $(x_i, y_i)$ in dataset $\mathcal{D}$ **do**
3:     $E_i \leftarrow$ Encoder($x_i$)          $\triangleright$ Pre-compute embedding
4: **end for**
5: Initialize an empty list $\mathcal{R}$ of results.
6: **for each** task $(P_t, x_t, y_t)$ in task list $\mathcal{T}$ **do**
7:     $C \leftarrow \varnothing$          $\triangleright$ Initialize the set of closest images
8:     **for each** possible label $y$ in $\mathcal{Y}$ **do**
9:         Find the $k$ closest embeddings to the task image $x_t$'s embedding, *i.e.*

10:         the unique indices $t_1, \dots, t_k$ that maximize

$$\arg\max_{t_1, \dots, t_k} \sum_{i=1}^{k} \text{Cosine Similarity}(E_t, E_{t_i})$$

such that $t_i \neq t$ and $y_{t_i} = y$ for all $i = 1, \dots, k$, where the cosine similarity is defined as

$$\text{Cosine Similarity}(a, b) = \frac{a \cdot b}{||a|| \, ||b||}.$$

11:         $C \leftarrow C\{(x_{t_i}, y)\}_{i=1}^{k}$          $\triangleright$ Store closest images and their labels
12:     **end for**
13:     Format the input $I$ to the LLM as follows:
14:     Include system prompt $P_{sys}$ describing the expected model's
15:     behavior.
16:     Include user prompt $P_t$ describing the setting and task.
17:     Interleave the example images and labels from $C$ represented as
18:         $\{(x, y) | x \in X, y \in Y\}$.
19:     **for each each** tuple $(x, y)$ in $C$ **do**
20:         $I \leftarrow I + (x, y)$          $\triangleright$ Append tuple to $I$
21:     **end for**
22:     Present the target image $x_t$.
23:     $R_t \leftarrow$ LLM($I$)          $\triangleright$ Invoke the *LLM* for this task
24:     $\mathcal{R} \leftarrow \mathcal{R} \cup \{(R_t, y_t)\}$          $\triangleright$ Store the result and ground truth label
25: **end for**

### kNN-based few-shot image sampling

The entire workflow is shown in detail as pseudocode in Box 1. Image feature vectors were created for each of the above-described datasets using the teacher backbone of the *'Phikon'* Vision Transformer (ViT-B 40 M Pancancer)[9] leading to a one- dimensional vector of length 768 for each image tile. During GPT-4V inference, for each test image $x$, the $k$ closest images of each possible target label $y$ were sampled for *kNN*-based in-context learning by measuring the cosine similarity in feature space. To prevent the model from learning patient-intrinsic morphologic tissue features as confounders to the desired label, we removed tile embeddings from the same patient if this information was available. Nevertheless, the main goal of our study is the comparison between GPT-4V in-context learning and training of specialized image classifiers. As outlined later, the comparisons are still valid in cases where overlap between test image and related patient tiles might occur, due to exactly matching in-context learning and training samples. The example images were included in the prompt in a way that the most similar images for each label were shown to the model first.

### Tile-level classification benchmarks

In this study, we first compared few-shot image in-context learning of GPT-4V with the performance of specialized computer vision models by training a classification layer atop four distinct models: *ResNet-18, ResNet-50, ViT-Tiny*, and *ViT-Small*. Each model has been initialized with ImageNet pretrained weights as a standard procedure[7,35]. Considering the relatively small test sample sizes in the experiments involving GPT-4V, we initially ensured a balanced comparison by training a newly initialized model on the identical set of *kNN*-sampled and normalized images for each test image across all datasets, leading to a total of 3600 trained models (Number of models × Total number of datasets × (Number of shots − zero shot) × Number of samples per dataset). Every training run was performed for ten epochs, employing the Adam Optimizer with a learning rate of 0.001, and using cross-entropy as the loss function. Next, we trained these models on the full training dataset using adaptive momentum optimization and learning rate scheduling for MHIST, PCAM, and CRC100K individually with 10% of the data as a validation set and checkpointing on validation accuracy. For comparison against histopathology foundation models, we either place a linear layer on top of the feature embeddings obtained from the Phikon or UNI model and train this layer for 1, 3, 5, and 10 epochs using the same hyperparameters as described above or compare the test sample features with the features, extracted from the training set by measuring cosine distance in representation space (nearest neighbor classification). We provide all training hyperparameters in Supplementary Table 4. Due to the balanced target label distribution, unweighted accuracy scores are reported for each of the models.

### Data analysis and visualization

For the generation of t-SNE, data was initially reduced to 200 principal components using Principal Component Analysis. The perplexity parameter was set at 30, and the process was initialized with a random seed of 42. All data visualization procedures were performed utilizing the Matplotlib and Seaborn packages.

### Reporting summary

Further information on research design is available in the Nature Portfolio Reporting Summary linked to this article.

## Data availability

All datasets used in this study are publically available and can be downloaded from https://huggingface.co/datasets/DykeF/NCTCRCHE100K (CRC100K), https://github.com/basveeling/pcam (PatchCamelyon) and https://bmirds.github.io/MHIST/ (MHIST). Source data are provided in this paper.

## Code availability

We provide all materials and code to reproduce and extend the analyses that were performed in this study upon publication under: https://github.com/Dyke-F/GPT-4V-In-Context-Learning.

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

## Acknowledgements

J.N.K. is supported by the German Federal Ministry of Health (DEEP LIVER, ZMVI1-2520DAT111; SWAG, 01KD2215B), the Max-Eder-Program of the German Cancer Aid (grant #70113864), the German Federal Ministry of Education and Research (PEARL, 01KD2104C; CAMINO, 01EO2101; SWAG, 01KD2215A; TRANSFORM LIVER, 031L0312A; TANGERINE, 01KT2302 through ERA-NET Transcan), the German Academic Exchange Service (SECAI, 57616814), the German Federal Joint Committee (Transplant.KI, 01VSF21048) the European Union's Horizon Europe and innovation program (ODELIA, 101057091; GENIAL, 101096312) and the National Institute for Health and Care Research (NIHR, NIHR213331) Leeds Biomedical Research Center. DT is funded by the German Federal Ministry of Education and Research (TRANSFORM LIVER, 031L0312A), the European Union's Horizon Europe and innovation program (ODELIA, 101057091), and the German Federal Ministry of Health (SWAG, 01KD2215B). G.W. is supported by Lothian NHS. The views expressed are those of the author(s) and not necessarily those of the NHS, the NIHR or the Department of Health and Social Care.

## Author contributions

D.F. designed and performed the experiments, evaluated and interpreted the results, and wrote the initial draft of the paper. G.W. provided scientific support for running the experiments and contributed to writing the paper. I.W., M.L., S.S., N.G.L., O.S.M.E.N., G.M.-F. contributed to writing the paper. DJ supervised the study. D.T. and J.N.K. designed and supervised the experiments and wrote the paper.

## Funding

## Competing interests

The authors declare the following competing interests. O.S.M.E.N. holds shares in StratifAI GmbH. J.N.K. declares consulting services for Owkin, France; DoMore Diagnostics, Norway; Panakeia, UK, and Scailyte, Basel, Switzerland; furthermore J.N.K. holds shares in Kather Consulting, Dresden, Germany; and StratifAI GmbH, Dresden, Germany, and has received honoraria for lectures and advisory board participation by AstraZeneca, Bayer, Eisai, MSD, BMS, Roche, Pfizer and Fresenius. D.T. received honoraria for lectures by Bayer and holds shares in StratifAI GmbH, Germany. The remaining authors declare no competing interests.
