## [Peer Review File · Nature Communications]

REVIEWER COMMENTS

Reviewer #2 (Remarks to the Author): Expert in AI for healthcare and large language models

This manuscript investigates the potential of multimodal large language models (LLM; also vision-language models, or VLM; GPT-4V in this study), to perform classification on medical images with in-context learning. In particular, prompting both via random additional images and kNN images was attempted on several histopathology classification datasets (CRC100K, PCam, MHIST), and compared against the results from specialized fine-tuned image classifiers. While timely, some issues might be considered:

1. In the Methods section, it is stated that "In the few-shot sampling prompts, we presented a sequence of k example images (where k equals 1, 3, 5 or 10), each followed by its corresponding label, in a repeated pattern: Specifically, we presented a single image corresponding to each label, cycling through the entire set of labels k times as further highlighted in Appendix B". It might be clarified as to whether this means that $k*c$ additional images are actually presented to the GPT-4V VLM - where c is the number of classes - for kNN few-shot in-context learning. Moreover, it might be clarified as to whether the same number of images (for identical k) were provided, for few shot random sampling.
2. Related to the above, Figure 1B and other references might be updated accordingly, since the figure currently suggests that k closest samples are presented for any particular k , in random/kNN learning/sampling (with results as reported in Figure 2).
3. Inspecting the few-shot prompts for the various datasets in Appendix C, it appears that the ground truth label of the additional images (whether for random/kNN learning) is not indicated in the prompt (Step 1). It might be clarified as to whether this is indeed the case, because if so, it would significantly broaden the applicability of the in-context learning technique in not requiring ground truth annotations (i.e. unsupervised learning).
4. In the Results section and corresponding Figure 2, the presentation of results on CRC100K (Fig 2A) and MHIST/PCAM (Fig 2B) appears different, in that it is unclear whether random or kNN in-context learning was applied to CRC100K, and there being no CI estimates for the random/kNN curves for MHIST/PCAM. It might be considered to standardize the presentation of the results for all three datasets, unless there is a good reason for the discrepancy.
5. In the Results section, it is then claimed that VLMs can achieve performance on par with retrained (specialized) vision transformers (i.e. ResNet-18/50, Tiny/Small-ViT, in Fig 3). However, from the statement that "To ensure a fair comparison, we train one distinct model for each target image shown to GPT-4V, with the identical images used for in-context learning as the training set" (and the Tile-Level Classification Benchmarks) suggests that this equivalent performance holds only when the narrow vision classifiers are also provided with extremely limited training data, which is largely to be expected.

The arguably more relevant comparison might be against specialized narrow vision classifiers trained on the actual full training data for each dataset, which would represent the performance of specialized

classifiers in practice, since actual implementations would be expected to use available labeled training data. For example, benchmark accuracy for PCam with basic specialized classifiers appears to be >93% (<https://www.kaggle.com/code/soumya044/histopathologic-cancer-detection/notebook>). That the comparison is only against specialized classifiers *trained under the same data training conditions* is not emphasized in the Abstract and Introduction, which might be considered to imply greater impact of the VLM than actually demonstrated.

6. Related to the above, Figure 2 suggests that performance might not have been maximized with $k=10$, for both kNN and random in-context learning. It might be considered to explore the effect of further increasing k , if feasible, and compare this against specialized classifiers at high k (i.e. developed with the actual full training set).

7. In the Image in-context learning improves text-based reasoning section, it is stated that Ada-002 and t-SNE were used for analyzing embeddings. The parameters (e.g. perplexity) used for these methods might be stated, possibly in supplementary material.

8. While it is then claimed that "The implementation of few-shot sampling, however, contributes to a more... pronounced separation of different labels within the embedding space... few-shot sampling assists the model to generate a consistent text-level reasoning trajectory to distinguish different targets", Figure 5 also shows that answer "labels" and label "labels" often do not correspond. This claim might thus be explained in greater detail.

9. It is then stated that Figure 5B shows an example where GPT-4V supposedly falsely classifies an image as tumor according to the original ground truth, only for further inspection to suggest that the image might actually contain tumor cells (i.e. the original ground truth was incorrect). This would appear to cast some doubt on the reliability of the other results (since they would be based on partially-incorrect ground truth). As such, was any effort taken to check the annotations for the three test sets used?

10. It is then stated that GPT-4V transfers knowledge from different domains (in particular the concept of "chicken wire") in making the right conclusions. It might be clarified whether "chicken wire" is actually an accepted/known description used in biomedical texts, for identifying tissue types.

11. In the Tile-Level Classification Benchmarks section, it is stated that for the various specialized classifiers, each training run was performed for ten epoches. It might be clarified as to how this number of epoches was chosen, since it is unclear whether convergence has been reached without a validation set in the training process. Moreover, full technical details (e.g. including learning rate decay) for these models might be included, possibly in supplementary material.

Reviewer #2 (Remarks on code availability):

Code review pending response to initial comments.

Reviewer #3 (Remarks to the Author): Expert in cancer digital pathology, machine learning, deep learning, breast cancer, and computer vision

Overall assessment

This work explores in-context learning as a data efficient approach to adapt vision-language foundation models to histopathology image classification without requiring model retraining. The main novelty of this work is that, to the best of my knowledge, it is the first to explore in-context learning to vision-language foundation models for medical image classification. Overall, the methodology is convincing, simple, and general. Another main strength of this paper is that the authors clearly describe how they have structured the input prompts and expected output format. This transparency is very important not only for reproducibility but also as the image classification performance could be sensitive to the structure and content of the zero/few-shot prompts. The results suggest competitive performance with other state-of-the-art approaches, although a more rigorous evaluation could entail fine-tuning the vision models on more data and weight initialization from foundation models pretrained on histopathology data instead of ImageNet. This would enable understanding the performance gap between in-context learning with the most robust methods for histopathology image analysis. Overall, the manuscript is very well written, and the results demonstrate the potential of in-context learning to adapt vision-language foundation models for medical tasks. This opens a new avenue for medical imaging research, enabling experts without technical knowledge on machine learning to adapt pretrained models to their clinical needs and without requiring extensive datasets.

Specific comments

Major comments:

a) 142-143 The authors state: “Next, we compare few-shot sampling with the current status quo in image classification, which involves retraining models from ImageNet weights.” This is correct for computer vision in general, but the current state-of-the-art in computational pathology is to fine-tune models pretrained on large-scale histopathology datasets. I suggest the authors perform additional experiments to compare few-shot classification using GPT-4V against several fine-tuned histopathology foundation models (e.g., PLIP, CONCH (vision/text), Phikon, CTransPath, Virchow, UNI (vision only))...

UNI: <https://www.nature.com/articles/s41591-024-02857-3>

CONCH: <https://www.nature.com/articles/s41591-024-02856-4>

PLIP: <https://www.nature.com/articles/s41591-023-02504-3>

b) The model outputs a score representing the model’s confidence about its decision. It would be valuable to the work to compare the “thoughts” outputs when the score is high and when the score is low, so the reader would know the reasons that lead the model to be confident (or not) about its prediction. It would also add value to the already done analysis when the model misclassifies a test sample.

c) Are the results strongly dependent on the prompt? Different users will end up using different prompts, even if a ‘guide’ is provided. It would be important to know the robustness of the result.

Minor comments:

a) Since the prompt development followed a trial-and-error approach, it would be good to present some of the strategies that did not work well. This will guide future work on prompt engineering to avoid some pitfalls.

b) Appendix D needs a caption and is not mentioned in the manuscript, so it is hard to know its purpose and importance.

c) lines 117-118 The authors state that: "This hypothesis has been shown with text-only tasks, but remains unclear for its application to biomedical images." But no references are included to support the statement. It would be relevant to cite works that use few-shot prompting to improve the performance of text-only foundation models.

d) lines 209 – The authors write: "Overall, these data show that vision language models hold great learning potential for medical image classification" From the current findings, one cannot be sure in-context learning generalizes to other medical imaging modalities. From the methodology and results, conclusions can only be made to histopathology data.

e) lines 240-241 The sentence states: "carefully selecting high-quality, few-shot examples can significantly enhance model performance." "high-quality examples" does not correctly describe what was done. At most, selecting semantically similar, few shot examples.

f) Line 251 Is this a limitation or a strength? 44 model calls seem a poor efficiency/performance trade off.

g) Lines 269-273 I suggest the authors strengthen the discussion regarding this work's limitations and future work. For instance, the current paper does not evaluate model robustness to the choice of user prompt. Future work could entail a comparison of model performance for different structures of the chain of thought (CoT) prompting.

Reviewer #4 (Remarks to the Author): Early Career Researcher co-reviewer

Reviewer #5 (Remarks to the Author): Early Career Researcher co-reviewer

I co-reviewed this manuscript with one of the reviewers who provided the listed reports as part of the Nature Communications initiative to facilitate training in peer review and appropriate recognition for co-reviewers.

Author Letter to the Reviewers

REVIEWER COMMENTS

Reviewer #2 (Remarks to the Author): Expert in AI for healthcare and large language models

This manuscript investigates the potential of multimodal large language models (LLM; also vision-language models, or VLM; GPT-4V in this study), to perform classification on medical images with in-context learning. In particular, prompting both via random additional images and kNN images was attempted on several histopathology classification datasets (CRC100K, PCam, MHIST), and compared against the results from specialized fine-tuned image classifiers. While timely, some issues might be considered:

1. In the Methods section, it is stated that "In the few-shot sampling prompts, we presented a sequence of k example images (where k equals 1, 3, 5 or 10), each followed by its corresponding label, in a repeated pattern: Specifically, we presented a single image corresponding to each label, cycling through the entire set of labels k times as further highlighted in Appendix B". It might be clarified as to whether this means that $k \cdot c$ additional images are actually presented to the GPT-4V VLM - where c is the number of classes - for kNN few-shot in-context learning. Moreover, it might be clarified as to whether the same number of images (for identical k) were provided, for few shot random sampling.

Author Answer: Thank you for highlighting this point. For each test image, we present k example images for each possible label y . This means, for instance in the setting of the CRC100K dataset that when we have $k=5$ samples per $y=8$ labels, this leads leads to $(k \cdot y) + 1 = 41$ (including the test image) images that are shown to GPT-4V every time. This setting is the same for random sampling and kNN-based sampling. We have adjusted the relevant part in the Methods section as follows: *"This means that for every test image, we present the model a total of $(k \cdot y) + 1$ images. This setting is the same for the kNN-based sample selection, which we further highlight below and in **Appendix B.**"* (p.10). We hope this improves clarity.

2. Related to the above, Figure 1B and other references might be updated accordingly, since the figure currently suggests that k closest samples are presented for any particular k , in random/kNN learning/sampling (with results as reported in Figure 2).

Author Answer: Thank you. We have updated Figure 1B for few shot random and kNN sampling accordingly, pointing out to readers that not k in total, but for each possible label y , k images (so $k \cdot y$ in total) were shown as examples to the model. Please let us know if this addresses your concerns.

3. Inspecting the few-shot prompts for the various datasets in Appendix C, it appears that the ground truth label of the additional images (whether for random/kNN learning) is not indicated in the prompt (Step 1). It might be clarified as to whether this is indeed

the case, because if so, it would significantly broaden the applicability of the in-context learning technique in not requiring ground truth annotations (i.e. unsupervised learning).

Author Answer: Thank you for spotting this issue. For few-shot learning, we have provided the correct label as outlined on p. 10 (Each image was prefaced with the phrase ‘*The following image contains*’). This indeed was not part of the “general prompt” as it had to happen dynamically at runtime (to catch the original label from the sampled examples for each dataset). This can be found in our codebase in the yaml configuration files (‘label_replacements’, i.e. here: <https://github.com/Dyke-F/GPT-4V>

-In-Context-

Learning/blob/4ffe61626b2f8a7d5540887451c79f886746aba1/config/CRC100K/binary/one_shot.yaml) and the GPT4EvalDataset in dataset.py (i.e. line 95).

Regarding this issue, we have improved our manuscript in the following ways: In the main text, we write “*Each image was prefaced with the phrase ‘The following image contains {y}’ for any possible label y.*” Moreover, we extended the few-shot prompt for all three datasets including the same template in the Supplementary Table B3.

4. In the Results section and corresponding Figure 2, the presentation of results on CRC100K (Fig 2A) and MHIST/PCAM (Fig 2B) appears different, in that it is unclear whether random or kNN in-context learning was applied to CRC100K, and there being no CI estimates for the random/kNN curves for MHIST/PCAM. It might be considered to standardize the presentation of the results for all three datasets, unless there is a good reason for the discrepancy.

Author Answer: Thank you for pointing this out. To test our hypothesis, we first ran an initial experiment with GPT-4V on a binary task on CRC100K to differentiate only between tumor (TUM) and non-tumor (NORM) tiles and using random sampling for in-context learning. This was considered as a simple proof-of-concept experiment. Simultaneously, similar to the idea that in text-based in-context learning we do not provide “random” few-shot examples, we sought to investigate the performance of providing similar images (in representation space) as examples. Our idea was fostered by the publication of the manuscript “*Can Generalist Foundation Models Outcompete Special-Purpose Tuning? Case Study in Medicine*”¹, where this kind of in-context learning was performed on text-based tasks and tested this idea on MHIST and PCAM. For readers to better follow the flow of our idea (showing that in-context learning works in general (Figure 2A) and can be improved by kNN-sampling (Figure 2B) we sought to present the data in this way.

To address your concerns, we have updated it as follows: We add clear comments in the figure caption (p. 15) to indicate when kNN- and when random-sampling was used. For readability, and as CIs are indicated in the below table, we initially left out CIs in Figure 2B, but have added them now to enhance consistency. Thank you for your feedback on this. We hope that this resolves your concerns.

5. In the Results section, it is then claimed that VLMs can achieve performance on par with retrained (specialized) vision transformers (i.e. ResNet-18/50, Tiny/Small-ViT, in Fig 3). However, from the statement that “To ensure a fair comparison, we train one distinct model for each target image shown to GPT-4V, with the identical images used for in-context learning as the training set” (and the Tile-Level Classification Benchmarks) suggests that this equivalent performance holds only when the narrow

vision classifiers are also provided with extremely limited training data, which is largely to be expected.

The arguably more relevant comparison might be against specialized narrow vision classifiers trained on the actual full training data for each dataset, which would represent the performance of specialized classifiers in practice, since actual implementations would be expected to use available labeled training data. For example, benchmark accuracy for PCam with basic specialized classifiers appears to be >93% (<https://www.kaggle.com/code/soumya044/histopathologic-cancer-detection/notebook>). That the comparison is only against specialized classifiers *trained under the same data training conditions* is not emphasized in the Abstract and Introduction, which might be considered to imply greater impact of the VLM than actually demonstrated.

Author Answer:

Thank you for your comment. We have now included benchmarks for comparing GPT-4V in-context learning with Resnet-18, Resnet-50, ViT-Tiny and ViT-Small, trained on the full training dataset for MHIST, PCAM and CRC100K. For training, we followed the examples from the link you provided above, training these models on the entire datasets each for one full epoch respectively including the hyperparameters as indicated. We have integrated the results into Figures 3 and 4 and the respective statistics tables. Please note, that our main message is the benefits of in-context learning regarding data- and resource efficiency and ease of use, and not to show how to outperform classical computer vision models. However, we can show that the performance gap between GPT-4V and vision models trained on tens of thousands of images for a specific task, can be substantially recovered by only a few samples for in-context learning. Additionally, in line with a major concern from Reviewer 3, we have added the results from comparing GPT-4V with in-context learning to foundation models (Phikon and UNI, which both represent the current state-of-the art in histopathology) here. We have therefore also updated the respective sections in the main text (“Vision-Language Models can achieve performance on par with retrained vision classifiers”, “In-context learning reduces the performance gap between generalist and histopathology foundation models” and “Discussion”). You can find the code here: https://github.com/Dyke-F/GPT-4V-In-Context-Learning/tree/train_classifiers

6. Related to the above, Figure 2 suggests that performance might not have been maximized with $k=10$, for both kNN and random in-context learning. It might be considered to explore the effect of further increasing k , if feasible, and compare this against specialized classifiers at high k (i.e. developed with the actual full training set).

Author Answer:

Thank you for this suggestion. Actually we had already tried it, but with 10 sample images per label for the binary tasks (PCAM and MHIST) and 5 for CRC100K are the maximum of samples we can currently successfully run via the API. When trying to scale these up in a reasonable manner (for instance 15 per class for PCAM, meaning 30+1 images in total per prompt) we unfortunately run into Rate-Limit Errors (“Too many Requests”) as shown below, so it is currently not possible to complete this idea even though we would have liked to perform an ablation study on the optimal number of top k samples. We will have to leave this out for exploration in the near future. We had stated this already in our manuscript - and now include the extension as to why we are

not able to increase the number of few-shot images: “Furthermore, our experiments have not indicated any saturation point in model efficacy when increasing the number of k-shot examples - however further increasing the number of sample images per task is not feasible due to exceeding the models capacity.” (p. 7-8).

```
[2024-04-24 20:14:24,830] [httpx][INFO] - HTTP Request: POST https://api.openai.com/v1/chat/completions "HTTP/1.1 429 Too Many Requests"
[2024-04-24 20:14:24,830] [openai_base_client][INFO] - Retrying request to /chat/completions in 0.944851 seconds
[2024-04-24 20:14:26,667] [httpx][INFO] - HTTP Request: POST https://api.openai.com/v1/chat/completions "HTTP/1.1 429 Too Many Requests"
[2024-04-24 20:14:26,669] [openai_base_client][INFO] - Retrying request to /chat/completions in 1.704771 seconds
[2024-04-24 20:14:29,335] [httpx][INFO] - HTTP Request: POST https://api.openai.com/v1/chat/completions "HTTP/1.1 429 Too Many Requests"
[2024-04-24 20:14:29,335] [openai_base_client][INFO] - Retrying request to /chat/completions in 1.704771 seconds
```

7. In the Image in-context learning improves text-based reasoning section, it is stated that Ada-002 and t-SNE were used for analyzing embeddings. The parameters (e.g. perplexity) used for these methods might be stated, possibly in supplementary material. **Author Answer:** Thank you for pointing this out. We added the respective (hyper-)parameters for the t-SNE into the section “Data Analysis and Visualization”. Ada-002 was the default embedding model at this time in order to convert text to numerical representations. There are no (hyper-)parameters to report.

8. While it is then claimed that "The implementation of few-shot sampling, however, contributes to a more... pronounced separation of different labels within the embedding space... few-shot sampling assists the model to generate a consistent text-level reasoning trajectory to distinguish different targets", Figure 5 also shows that answer "labels" and label "labels" often do not correspond. This claim might thus be explained in greater detail.

Author Answer: Thank you for your comments. Our approach in this study involves the use of text embedding models to convert textual data into semantic representations. Essentially, texts conveying similar content (e.g., descriptions of tumor tissue) are represented closely in semantic space, whereas those describing dissimilar content (such as normal tissue patterns or other types of tissue) are distinctly separated.

In our research, we specifically focus on embedding the Chain-of-Thought reasoning executed by the model before it makes a decision. This method differs from direct-shot predictions, which we did not explore due to the substantial evidence supporting the effectiveness of Chain-of-Thought reasoning in enhancing the performance of large language models across various domains.

Regarding the visualizations presented in Figure 5, we employed t-SNE to depict these embeddings. In the zero-shot scenario, whether we compare the model's reasoning to its answers or to the ground truth labels, the embeddings generally form only two distinct clusters, indicating that there is no clear pattern in the model's reasoning to separate between the 8 available classes.

In settings where few-shot learning is applied (five shots, with less pronounced effects in one- and three-shot experiments), we observe a clearer distinction with up to six or seven visually distinct clusters. These clusters also show greater homogeneity with respect to the corresponding ground truth labels, such as the label "NORM". This is consistent with the findings from Figure 4B, although some overlap persists (notably between categories like Tumor and Debris), suggesting areas for further investigation. We believe these findings demonstrate that supplying the model with sample images enables it to distinguish between different histological patterns at the text level before making a decision, which in turn is associated with enhanced accuracy, as evidenced by our other experiments. This hypothesis is supported by increased silhouette scores

that measure how close a datapoint is related to its “ground truth” (which is either the model’s answer or the real label). We see a steady increase in silhouette scores when we increase the number of few-shot samples. We adapted Figure 5A and the main text (p.5-6) with further explanations.

9. It is then stated that Figure 5B shows an example where GPT-4V supposedly falsely classifies an image as tumor according to the original ground truth, only for further inspection to suggest that the image might actually contain tumor cells (i.e. the original ground truth was incorrect). This would appear to cast some doubt on the reliability of the other results (since they would be based on partially-incorrect ground truth). As such, was any effort taken to check the annotations for the three test sets used?

Author Answer:

Thank you for highlighting this. Our goal in this manuscript is to convey the benefits in favour of in-context learning regarding compute- and data efficiency as well as convenience of use for researchers from outside the computer science fields and also offers the benefits of providing explanations in natural language. We did not manually re-check all ground truth annotations, as this would require us for the sake of completeness to also check the training datasets, which are relatively large, i.e. 100k samples for CRC100K. If more “erroneous” responses were to be identified, this would even mean that GPT-4s performance would even be underestimated. We will aim to further explore this research area in the upcoming months.

10. It is then stated that GPT-4V transfers knowledge from different domains (in particular the concept of “chicken wire”) in making the right conclusions. It might be clarified whether “chicken wire” is actually an accepted/known description used in biomedical texts, for identifying tissue types.

Author Response: In our investigation, we observed that GPT-4, accessed either via the API or ChatGPT (images included below), does not recognize the term “Chicken-Wire pattern” as a formal pathological descriptor for the appearance of “healthy” adipose tissue. Instead, this term is consistently associated solely with capillary formations in the context of myxoid liposarcoma. We performed a manual search, which supported the absence of this terminology in reference to normal adipose tissue². This discrepancy underscores GPT-4’s capacity for abstract concept recognition and transfer, as it has the ability to apply the visual pattern of chicken wire to the adipose structures observed in this specific image. This capability offers potential in fields focused on explainability and education. To address your issues, we have addressed this point on p. 6 (*According to existing literature², the term “chicken wire pattern” is established within the domain of pathology, yet only regarding the appearance of liposarcomas and other malignancies related to fat tissue. However, it appears not common to describe the architecture of normal, healthy adipose tissue. The capability of GPT-4 to transfer its understanding of the physical appearance of chicken wire to the shape of adipose tissue in histopathology demonstrates the ability for transfer learning and holds potential in areas like AI explainability and teaching.*)

 You

What is a chicken wire pattern in fat cell histology?

 ChatGPT

In fat cell histology, the "chicken wire" pattern refers specifically to the appearance of the capillary network within certain types of lipomas or liposarcomas, particularly those with a myxoid matrix. This pattern is characterized by very fine, regularly spaced capillaries that branch out to form a network that closely resembles chicken wire fencing. This distinct vascular pattern is significant in the pathological evaluation as it is a key diagnostic feature, especially in distinguishing myxoid liposarcomas from other types of soft tissue tumors. The presence of this pattern can indicate a specific behavior and origin of the tumor, aiding in the proper management and treatment planning for the patient.

11. In the Tile-Level Classification Benchmarks section, it is stated that for the various specialized classifiers, each training run was performed for ten epoches. It might be clarified as to how this number of epoches was chosen, since it is unclear whether convergence has been reached without a validation set in the training process. Moreover, full technical details (e.g. including learning rate decay) for these models might be included, possibly in supplementary material.

Author Response:

Thank you for your insightful comments. In our initial analysis, we limited our comparison to the 1-10 images used for in-context learning in GPT-4V, without incorporating learning-rate scheduling or other training optimizations such as validation sets. This decision was based on the practicality of processing the entire training set in a single batch per epoch, a condition under which traditional optimization techniques generally show minimal benefit, especially with limited data. However, acknowledging the need for a more rigorous evaluation, we have now expanded our study to include retraining all four models on their complete datasets. We have also made comparisons with state-of-the-art foundation models, such as UNI and Phikon. To address your concerns comprehensively, we have conducted additional training on all data, incorporating a 10% validation set. All pertinent technical details, including learning rates and decay parameters, have been summarized in Supplementary Material 4. We hope that these enhancements and clarifications will satisfactorily resolve the issues raised.

Reviewer #3 (Remarks to the Author): Expert in cancer digital pathology, machine learning, deep learning, breast cancer, and computer vision

Overall assessment

This work explores in-context learning as a data efficient approach to adapt vision-language foundation models to histopathology image classification without requiring model retraining. The main novelty of this work is that, to the best of my knowledge, it is the first to explore in-context learning to vision-language foundation models for medical image classification. Overall, the methodology is convincing, simple, and general. Another main strength of this paper is that the authors clearly describe how they have structured the input prompts and expected output format. This transparency is very important not only for reproducibility but also as the image classification performance could be sensitive to the structure and content of the zero/few-shot

prompts. The results suggest competitive performance with other state-of-the-art approaches, although a more rigorous evaluation could entail fine-tuning the vision models on more data and weight initialization from foundation models pretrained on histopathology data instead of ImageNet. This would enable understanding the performance gap between in-context learning with the most robust methods for histopathology image analysis. Overall, the manuscript is very well written, and the results demonstrate the potential of in-context learning to adapt vision-language foundation models for medical tasks. This opens a new avenue for medical imaging research, enabling experts without technical knowledge on machine learning to adapt pretrained models to their clinical needs and without requiring extensive datasets.

Specific comments

Major comments:

a) 142-143 The authors state: "Next, we compare few-shot sampling with the current status quo in image classification, which involves retraining models from ImageNet weights." This is correct for computer vision in general, but the current state-of-the-art in computational pathology is to fine-tune models pretrained on large-scale histopathology datasets. I suggest the authors perform additional experiments to compare few-shot classification using GPT-4V against several fine-tuned histopathology foundation models (e.g., PLIP, CONCH (vision/text), Phikon, CTransPath, Virchow, UNI (vision only))...

UNI: <https://www.nature.com/articles/s41591-024-02857-3>

CONCH: <https://www.nature.com/articles/s41591-024-02856-4>

PLIP: <https://www.nature.com/articles/s41591-023-02504-3>

Author Response:

Thank you for your response. Indeed, we acknowledge that we did not compare to the current gold standard foundation models in histopathology.

To address your concern we have made the following adjustments to our manuscript:

1. We have added experiments where we compare GPT-4V k-shot in-context learning with the performance of the Vision Classifiers (Resnet18, Resnet50, Vit-Tiny and Vit-Small), trained on the respective full datasets. We have added these as a baseline to Figures 3 and 4.
2. Additionally, we have added results from experiments of using Phikon and UNI which represent the two state-of-the-art models with the best available performance for histopathology. We include results from respective experiments from a) training a linear layer on top of both of these models for 1, 3, 5 and 10 epochs on all images from each dataset and validate the performance on the same samples as for the in-context learning in GPT-4 and b) using kNN classification by using the top 1 most similar feature-embedding from the training set to determine the models response on the test samples. In both cases, we can show that in-context learning can drastically reduce the performance gap between a generalist and a histopathology foundation model with only a handful of images in a resource-efficient manner.
3. Please note that at the time we performed the experiments (November 2023) and submitted the manuscript (early January 2024) UNI was not available yet. Also, unfortunately Virchow is a proprietary model (Paige) that we have no

access to. As Phikon and especially UNI as of today represent the state-of-the-art models, we are optimistic that these experiments are representative enough to convey our message.

4. Overall, please also note that our main goal is not to achieve state-of-the-art performance, but to showcase that in-context learning with multimodal foundation models might hold advantages, such as being a data- and resource-efficient approach for histopathology image classification. Another advantage is that the ease of using in-context learning might also democratize access to researchers without extensive experience in computer sciences. We therefore have not taken into account extensive hyperparameter tuning and other optimization techniques to achieve the maximum possible performance per model, but have initialized all hyperparameters with reasonable defaults.
5. The respective code can be found here: https://github.com/Dyke-F/GPT-4V-In-Context-Learning/tree/train_classifiers.

b) The model outputs a score representing the model's confidence about its decision. It would be valuable to the work to compare the "thoughts" outputs when the score is high and when the score is low, so the reader would know the reasons that lead the model to be confident (or not) about its prediction. It would also add value to the already done analysis when the model misclassifies a test sample.

Author Response:

We did not further investigate a comparison between the score and the model's response, as this would require a drastically larger sample size to enable quantitative evaluation. Moreover, to date several, ideally much more reliable methods to measure and quantify model "confidence" exist, for instance the use of log-probs per each token. We would suggest leaving this exploration for another project in the future in a more sophisticated setting.

c) Are the results strongly dependent on the prompt? Different users will end up using different prompts, even if a 'guide' is provided. It would be important to know the robustness of the result.

Author Response:

Thank you for highlighting this. We have initially given the model only a very generic prompt on a held-out development set (akin to "You are a pathology assistant. Classify this image given the following options ..."). This works out of the box, but had 3 drawbacks at the time: 1. OpenAI guardrails including model refusals ("I'm sorry I cannot help. You should consult a healthcare professional."). To circumvent this, we had to frame the setting as a "hypothetical scenario" and included few-shot examples of undesired model outputs. 2. To scale the evaluations, we needed consistent outputs (for instance in json format), so we provided a json template. 3. In our initial evaluations, we saw that the model would almost always choose to classify an image as tumor if there were tumor cells, despite being necrotic. Therefore we have added task-specific guardrails. Additionally we prompted the model to perform a reasoning step similar to Chain-of-Thought which is a standard procedure.

Also, as you have suggested in comment g), we have included the role of future work in prompt engineering in the discussion.

Minor comments:

a) Since the prompt development followed a trial-and-error approach, it would be good to present some of the strategies that did not work well. This will guide future work on prompt engineering to avoid some pitfalls.

Author Response:

Thank you for highlighting this. Please see attached our answer to the above comment (c) and the Section “Prompting and random few-shot image in-context learning” where we describe in detail the pitfalls we encountered during prompt development and how we could iteratively resolve them.

b) Appendix D needs a caption and is not mentioned in the manuscript, so it is hard to know its purpose and importance.

Author Response:

Thank you for your observations. To clarify, Appendix D features word clouds that depict the most frequent words in the models' responses, serving solely for illustrative purposes. This appendix is intended as a supplementary visualization to the embeddings presented in Figure 5. It demonstrates that the models' vocabulary aligns closely with the accurate clinical diagnoses. For example, the predominant terms for "normal lymph node tissue" in the PCAM dataset are "lymph node", whereas for lymph node metastases, the terms expand to include "lymph node" alongside "metastatic" and "breast cancer". This visual representation is included as a complementary reference to Figure 5, enhancing the interpretability of our findings. We have updated Figure 5 with a reference to Appendix D.

c) lines 117-118 The authors state that: “This hypothesis has been shown with text-only tasks, but remains unclear for its application to biomedical images.” But no references are included to support the statement. It would be relevant to cite works that use few-shot prompting to improve the performance of text-only foundation models.

Author Response:

Thank you. The landmark paper for few-shot prompting is the GPT-3 paper. For images, the pioneering paper on Vision-Language Models including performance gains when using in-context learning is “Flamingo: a Visual Language Model for Few-Shot Learning”³ (Figure 2, p. 3), which however does not cover biomedical images. We have included these as a reference to the above statement.

d) lines 209 – The authors write: “Overall, these data show that vision language models hold great learning potential for medical image classification” From the current findings, one cannot be sure in-context learning generalizes to other medical imaging modalities. From the methodology and results, conclusions can only be made to histopathology data.

Author Response:

This is true, thank you for pointing this out. We have rephrased the above sentence to “Overall, these data demonstrate that vision language models possess substantial potential for medical image classification in histopathology, ...”

e) lines 240-241 The sentence states: “carefully selecting high-quality, few-shot examples can significantly enhance model performance.” “high-quality examples” does not correctly describe what was done. At most, selecting semantically similar, few shot examples.

Author Response:

Thank you again for finding this error. We have rephrased the sentence to: “*Additionally, our results indicate that deliberately selecting few-shot examples, which are semantically similar to the test image, can substantially improve the performance of the model.*” We agree that this is a much better description.

f) Line 251 Is this a limitation or a strength? 44 model calls seem a poor efficiency/performance trade off.

Author Response: This is definitely a strong limitation, as this approach is currently not scalable and compute- and also financially expensive. This is why - including given the preview-status of the GPT-4V API that only permitted a handful of images per day and user when conducting the experiments - we refrained from exploring ensembling methods. Nevertheless, these techniques have attracted some attention across the field recently, for instance in this paper⁴, which showed that applying a majority vote over multiple different calls to GPT-3.5 drastically improved performance (Note that this was on text cases only). We would leave further explorations of enhancing our method (with model ensembling) for future projects. In the main text, we have also slightly modified the sentence to convey the message of a poor efficiency/performance trade off for the 44 model calls. Thank you for highlighting this.

g) Lines 269-273 I suggest the authors strengthen the discussion regarding this work’s limitations and future work. For instance, the current paper does not evaluate model robustness to the choice of user prompt. Future work could entail a comparison of model performance for different structures of the chain of thought (CoT) prompting.

Author Response:

Thanks, we have included this aspect in our discussion (p. 7), highlighting several shortcomings of our work. However, we believe that with the current paradigm of models becoming “smarter” in the near future, the relevance of choosing the right prompt might decrease.

References for the Response Letter only

1. Nori, H. *et al.* Can Generalist Foundation Models Outcompete Special-Purpose Tuning? Case Study in Medicine. *arXiv [cs.CL]* (2023).
2. Vimal, M. & Nishanthi, A. Food eponyms in Pathology. *J. Clin. Diagn. Res.* **11**, EE01 (2017).
3. Alayrac, J.-B. *et al.* Flamingo: a Visual Language Model for Few-Shot Learning. *arXiv [cs.CV]* (2022).
4. Li, J., Zhang, Q., Yu, Y., Fu, Q. & Ye, D. More Agents Is All You Need. *arXiv [cs.CL]* (2024).

REVIEWERS' COMMENTS

Reviewer #1 (Remarks to the Author):

Reviewer #1 (Remarks on code availability):

N/A

Reviewer #2 (Remarks to the Author):

We thank the authors for largely addressing our previous comments, in particular the additional experiments performed for specialized classifiers.

The remaining concern would be on Point 9, on possible errors in the test set ground truth. While it was stated that "We did not manually re-check all ground truth annotations, as this would require us for the sake of completeness to also check the training datasets", there is to the best of our knowledge no necessity to recheck training set ground truth, especially as deep neural network classifiers are fairly robust to training set errors (see e.g. "Impact of Label Noise on the Learning Based Models for a Binary Classification of Physiological Signal", Ding et al., Sensors 2022 Oct; 22(19): 7166). As such, it might be considered to minimally perform a brief check to estimate the amount of error present in the test data, possibly from subsample(s).

Reviewer #2 (Remarks on code availability):

Reviewer #3 (Remarks to the Author):

Overall, the authors have responded positively to the comments/suggestions, so that the article is much clearer and the main message is emphasised. The new version addresses some concerns we had about the sensitivity and robustness of the proposed approach. The main problem of the original article, which concerned the comparison between the proposed methodology and the state of the art, seems to me to have been resolved.

Despite the performance gap compared to foundation models, the methodology is a pioneering work in histopathology/medical imaging and has some advantages.

I believe that all doubts and suggestions have been sufficiently addressed by the authors in this new version of the document.

Reviewer #4 (Remarks to the Author):

Reviewer #5 (Remarks to the Author):

Authors Response:

Reviewer #1 (Remarks to the Author):

Author's Comment: Thank you very much.

Reviewer #1 (Remarks on code availability):

N/A

Reviewer #2 (Remarks to the Author):

We thank the authors for largely addressing our previous comments, in particular the additional experiments performed for specialized classifiers.

The remaining concern would be on Point 9, on possible errors in the test set ground truth. While it was stated that "We did not manually re-check all ground truth annotations, as this would require us for the sake of completeness to also check the training datasets", there is to the best of our knowledge no necessity to recheck training set ground truth, especially as deep neural network classifiers are fairly robust to training set errors (see e.g. "Impact of Label Noise on the Learning Based Models for a Binary Classification of Physiological Signal", Ding et al., Sensors 2022 Oct; 22(19): 7166). As such, it might be considered to minimally perform a brief check to estimate the amount of error present in the test data, possibly from subsample(s).

Author's Comment: Thank you very much for your detailed feedback.

We have now performed a brief estimate of the error rate in the CRC 100K dataset. We have sampled 25 images from each of the 9 classes and asked 3 human evaluators to re-classify them into one of the 9 groups independently. Below we show that there was a 100% agreement for the classes MUC, DEB, ADI, TUM, LYM, MUS and BACK and only minimal disagreement on the classes NORM, MUS and STR. The latter is likely due to phenotypic overlaps between desmoplastic stroma or invasion of tumour cells into the surrounding tissue. In the plot below, "Class: ..." denotes the "original" label in the CRC100K dataset, while the x-axis labels show the distribution of labels, assigned by the human evaluators. We hope this addresses your concerns.

Reviewer #2 (Remarks on code availability):

Author's Comment: Thank you very much for your work and feedback on our manuscript.

Reviewer #3 (Remarks to the Author):

Overall, the authors have responded positively to the comments/suggestions, so that the article is much clearer and the main message is emphasised. The new version addresses some concerns we had about the sensitivity and robustness of the proposed approach.

The main problem of the original article, which concerned the comparison between the proposed methodology and the state of the art, seems to me to have been resolved.

Despite the performance gap compared to foundation models, the methodology is a pioneering work in histopathology/medical imaging and has some advantages.

I believe that all doubts and suggestions have been sufficiently addressed by the authors in this new version of the document.

Author's Comment: Thank you so much for your feedback and help to make our article more robust. We hope we have addressed all your concerns.

Reviewer #4 (Remarks to the Author):

Author's Comment: Thank you very much for your support and help in improving our manuscript.

Reviewer #5 (Remarks to the Author):

Author's Comment: Thank you very much for your support and help in improving our manuscript.